# Independent Associated SNPs at SORCS3 and Its Protein Interactors for Multiple Brain-Related Disorders and Traits

**DOI:** 10.3390/genes14020482

**Published:** 2023-02-14

**Authors:** Muhammad Kamran, Aodán Laighneach, Farhana Bibi, Gary Donohoe, Naveed Ahmed, Asim Ur Rehman, Derek W. Morris

**Affiliations:** 1Department of Pharmacy, Faculty of Biological Sciences, Quaid-i-Azam University, Islamabad 45320, Pakistan; 2Centre for Neuroimaging, Cognition and Genomics (NICOG), School of Biological and Chemical Sciences and School of Psychology, University of Galway, H91 CF50 Galway, Ireland; 3Department of Biosciences, Grand Asian University, Sialkot 51040, Pakistan

**Keywords:** SORCS3, association, GWAS, brain disorders, gene expression

## Abstract

Sortilin-related vacuolar protein sorting 10 (*VPS10*) domain containing receptor 3 (*SORCS3*) is a neuron-specific transmembrane protein involved in the trafficking of proteins between intracellular vesicles and the plasma membrane. Genetic variation at *SORCS3* is associated with multiple neuropsychiatric disorders and behavioural phenotypes. Here, we undertake a systematic search of published genome-wide association studies to identify and catalogue associations between *SORCS3* and brain-related disorders and traits. We also generate a *SORCS3* gene-set based on protein–protein interactions and investigate the contribution of this gene-set to the heritability of these phenotypes and its overlap with synaptic biology. Analysis of association signals at *SORSC3* showed individual SNPs to be associated with multiple neuropsychiatric and neurodevelopmental brain-related disorders and traits that have an impact on the experience of feeling, emotion or mood or cognitive function, while multiple LD-independent SNPs were associated with the same phenotypes. Across these SNPs, alleles associated with the more favourable outcomes for each phenotype (e.g., decreased risk of neuropsychiatric illness) were associated with increased expression of the *SORCS3* gene. The *SORCS3* gene-set was enriched for heritability contributing to schizophrenia (SCZ), bipolar disorder (BPD), intelligence (IQ) and education attainment (EA). Eleven genes from the *SORCS3* gene-set were associated with more than one of these phenotypes at the genome-wide level, with *RBFOX1* associated with SCZ, IQ and EA. Functional annotation revealed that the *SORCS3* gene-set is enriched for multiple ontologies related to the structure and function of synapses. Overall, we find many independent association signals at *SORCS3* with brain-related disorders and traits, with the effect possibly mediated by reduced gene expression, resulting in a negative impact on synaptic function.

## 1. Introduction

Sortilin-related vacuolar protein sorting 10 (VPS10) domain containing receptor 3 (*SORCS3*) is a member of the type-I transmembrane receptor family and the VPS10 protein family. Other members of this family include *SORCS2*, *SORCS1*, sortilin (*SORT*) and sortilin-related receptor 1 (*SORL1*) [1]. Proteins encoded by this gene family share a common VPS10 domain at the N-terminus. A consensus motif for proprotein convertase processing is present at the N-terminal segment of this domain, while ten conserved cysteine residues contribute to the C-terminal segment of the domain [2,3]. There is also a leucine-rich segment, a transmembrane and a short C-terminal cytoplasmic domain that interacts with the adaptor molecules following the VPS10 domain [4]. These proteins are expressed at high levels in the central nervous system and at lower levels in other tissues and cells, such as the adrenal gland, thyroid, B-lymphocytes, skeletal muscle and heart [5].

VPS10 domain receptors are sorting receptors that are involved in controlling the intracellular trafficking of target proteins in neurons [6]. It facilitates the transport of proteins between the Golgi apparatus, endosomes, lysosomes, secretory granules and the plasma membrane. It also has a role in signal transduction that signifies the participation of *SORT* in different biological pathways [4,7]. By attenuating BDNF signalling, both *SORCS1* and *SORCS3* can control the energy balance and production of orexigenic peptides [8]. *SORCS1* acts to regulate synaptic transmission and plasticity because, when compared to GABAergic signalling, it strongly regulates the functions of glutamate receptors [9]. SORCS3 is also involved in the binding of nerve growth factor (NGF) and platelet-derived growth factor (PDGF-BB) [10,11]. *SORCS3* gene expression occurs exclusively in the nervous system and is limited to vesicular components only. Neuronal activity induces the hippocampal expression of *SORCS3* [12,13]. *SORCS3* also affects NDMA receptor-dependent and -independent long-term depression; hence, it acts as a postsynaptic modulator of synaptic depression and fear extension [14].

The *SORCS3* gene maps to chromosome 10q23.3 and encodes a protein composed of 1297 amino acids [15]. Studies have identified associations between both rare and common variants of the *SORCS3* gene and multiple psychiatric disorders, neurodegenerative disorders and other brain-related disorders and phenotypes. A homozygous missense mutation [NM_014978.2; c. 3110C > G (p.Thr1037Ser)] in *SORCS3* was reported in two brothers with infantile spasm and intellectual disability, which supports the crucial role of this gene in the central nervous system. Born to consanguineous parents, the two affected brothers presented with global developmental delay, infantile spasm, intellectual disability, hypotonia and microcephaly. Fibroblasts of the patients were reported to be structurally different, having less proliferation ability and viability than the normal fibroblasts [16]. A search for identical-by-descent shared segments in cases of multiple sclerosis (MS) and controls from the isolated Faroe Island population identified a haplotype with significantly higher frequency in cases that spanned the entire *SORCS3* gene [17]. The Autism Sequencing Consortium reported a de novo nonsense variant in this gene in an autism spectrum disorder (ASD) proband [18].

However, it has been the application of genome-wide association studies (GWAS) to brain-related disorders and traits that has resulted in a remarkable number of associations with *SORCS3*. *SORCS3* has been associated with major depressive disorder (MDD) [19], bipolar disorder (BPD) [20], schizophrenia (SCZ) [21], attention deficit hyperactivity disorder (ADHD) [22], ASD [23], dementia in women [24], neuroticism [25,26], educational attainment [27] and intelligence (IQ) [22,25,27,28,29]. In a multi-trait GWAS analysis of SCZ, BPD, MDD, ASD and ADHD, *SORCS3* was the only gene associated with all five disorders [23].

As these studies have identified up to hundreds of loci containing thousands of genes, only a small proportion of associated genes are highlighted and discussed in the manuscripts describing these studies. Therefore, studies may only be reporting genetic associations at *SORCS3* in their Appendix A. Where *SORCS3* does feature in main texts, there has been little data supplied on the location of associated variants, their possible function and their linkage disequilibrium (LD) relationship to other genetic variants that have been reported at the gene in other GWAS.

Here, we undertake a systematic search of published GWAS to identify and catalogue associations between *SORCS3* and brain-related disorders and traits. In this study, we consider brain-related disorders that are neuropsychiatric or neurodevelopmental in nature and traits that have an impact on the experience of feeling, emotion or mood or cognitive function. We identify multiple LD-independent SNPs associated with different phenotypes and explore the putative functional effect of these variants on gene expression. We build a *SORCS3* gene-set based on protein–protein interaction data. We test for a contribution of this gene-set to brain-related disorders and traits and for the enrichment of this gene-set in functional elements of the synapse.

## 2. Materials and Methods

### 2.1. Ethics Statement

Data were directly downloaded from published studies, and no additional ethics approval was needed. Each study is referenced, and details on ethics approval are available in each manuscript.

### 2.2. Sourcing of GWAS Results, LD Information and eQTL Data

GWAS results were sourced from the NHGRI-EBI GWAS Catalog (https://www.ebi.ac.uk/gwas/home) (accessed on 31 May 2022) [30]. The LDlink (https://ldlink.nci.nih.gov/) (accessed on 6 July 2022) [31] suite of web-based applications was used to interrogate linkage disequilibrium (LD) based on the CEU reference panel (Utah residents from north and west Europe) sub-population of the 1000 Genomes Project (1000 G) using genome build GRCh38 coordinates for SNP and gene locations. The HaploReg (https://pubs.broadinstitute.org/mammals/haploreg/haploreg.php) (accessed on 6 July 2022) tool was used for exploring annotations of SNPs in haplotype blocks, with LD calculated using the 1000 G Phase 1 European population data [32]. Expression quantitative trait loci data (eQTL) were sourced from the Genotype-Tissue Expression (GTEx) project (https://gtexportal.org) (accessed on 12 June 2022) [33].

### 2.3. GWAS Data

Non-multi-trait phenotypes with at least one SNP–phenotype association at SORCS3 at genome-wide significant levels with available summary statistics (n = 13) were considered for heritability enrichment analysis. GWAS summary statistics results were sourced for SCZ [34], IQ [28], EA [27], BPD [35], MDD [36], ASD [37], ADHD [22], childhood maltreatment [38], feeling miserable [39], feeling nervous [39], risk-taking [40], neuroticism [41] and mood instability [42]. In addition to the test phenotypes, five control phenotypes were also used: Alzheimer’s disease (AlzD) [43], stroke [44], coronary artery disease (CAD) [45], Crohn’s disease [46] and type 2 diabetes (T2D) [47].

### 2.4. SORCS3 Gene-Set

The SORCS3 gene-set was generated by building a network of genes around SORCS3 based on protein–protein interaction data from the following databases: STRING (https://string-db.org/) (accessed on 7 December 2021), BioGRID (https://thebiogrid.org/) (accessed on 7 December 2021), IntACT Molecular Interaction Database (https://www.ebi.ac.uk/intact/home) (accessed on 7 December 2021) and Human Protein Reference Database (HPRD: http://hprd.org/index_html) (accessed on 7 December 2021). First, using default settings, genes listed as direct (1st degree, n = 14) or indirect (2nd and 3rd degree; n = 19) interactors of SORCS3 were found using these databases. We then used each of the fourteen 1st degree protein interactors as bait to search for their interactors in each database.

### 2.5. Stratified Linkage Disequilibrium Score Regression (sLDSC) Analysis

To investigate if the SORCS3 gene interaction network was enriched for heritability contributing to the phenotypes highlighted by individual association signals, stratified linkage disequilibrium score regression (sLDSC) (https://github.com/bulik/ldsc) (accessed on 28 June 2022) [48,49] was performed. The start and stop coordinates of each of the 412 SORCS3 protein-interacting genes on GRCh37 were found using biomaRt [50]. HapMap Project phase 3 SNPs with a MAF > 0.05 within a 100 kb upstream and downstream window [51] of these regions were considered in this analysis. LD scores between SNPs within a 1cM window were estimated using the 1000 Genomes Phase 3 European reference panel. SNP heritability for each phenotype was stratified for the SORCS3 gene network using a model accounting for heritability associated with 53 functional genomic annotations found in the baseline model [49]. Enrichment for heritability compared to the baseline model was calculated with a corresponding *p*-value. Enrichments surviving a Bonferroni-corrected *p*-value of <0.05 were considered significantly enriched.

### 2.6. MAGMA Gene-Set Analysis

We used MAGMA v1.09 [52] to generate gene-based p-values from summary statistics from various GWAS. The MAGMA analysis involved two steps. First, in the annotation step, we mapped SNPs with available GWAS results onto protein-coding genes (GRCh37/hg19 start-stop coordinates ±20 kb). Second, in the gene analysis step, we computed gene p-values for each GWAS dataset. This gene analysis is based on a multiple linear principal components regression model that accounts for LD between SNPs. The European panel of the 1000 G data was used as a reference panel for LD. Genes surviving a Bonferroni-corrected *p* < 0.05 were considered genome-wide significantly associated with each respective phenotype.

### 2.7. Functional Annotation

SynGO (https://syngoportal.org/) (accessed on 20 January 2022) was used to perform an overrepresentation analysis of the SORCS3 gene-set using an evidence-based, expert-curated resource of gene ontology terms for synapse function [53]. A “brain expressed” background set of genes was selected for gene-set enrichment analysis (GSEA); it contains 18,035 unique genes in total, of which 1225 overlap with SynGO annotated genes. The GSEA only tested SynGO terms with at least three matching input genes.

## 3. Results

### 3.1. Association Signals at SORCS3 in GWAS Data

A review of the NHGRI-EBI GWAS Catalog identified 72 SNP–phenotype associations at *SORCS3* at genome-wide significant levels (*p* < 5 × 10^−8^) between 47 different SNPs and 25 different neuropsychiatric disorders or behavioural or brain-related phenotypes (Appendix A). The catalogued studies included analyses of individual disorders and traits (e.g., *schizophrenia*, *intelligence*) as well as multi-phenotype studies (e.g., a combination of *unipolar depression and bipolar disorder*). Associated SNPs are located across the full length of the gene. Fifteen of the forty-seven SNPs are associated with more than one phenotype. For example, rs11599236 (intron 1) is individually associated with *feeling miserable*, *mood instability*, *educational attainment*, *unipolar depression*, a combination of *unipolar depression and mood disorder* and the cross-disorder combination of *ADHD, ASD, SCZ, BPD and MDD* (Figure 1).

To further study the inter-relationship between the association signals, we explored the LD relationships between 46 of the 47 SNPs (rs7906899 was missing from 1000 G GRCh38 data). This identified two large blocks of LD across the gene (Figure 2). The first extends from the 5′ end of the gene to intron 14, and the second extends from intron 14 to the 3′ end of the gene. We LD-pruned (*r*^2^ < 0.1) this set of 46 SNPs to identify a set of 8 SNPs that tag all association signals (Appendix A). Three SNPs (rs11192147, rs12416372 and rs11596241) tag 35 of the 46 SNPs and each of the three LD-dependent groups of SNPs are associated with a broad range of phenotypes. Of the remaining five tag SNPs, rs790647 tags a group of 5 SNPs associated with cognitive phenotypes and rs12356045 tags a group of 3 SNPs associated with cognitive phenotypes and a combination of *ADHD, substance abuse and antisocial behaviour*. The remaining three tag SNPs that are not in LD with other SNPs; rs10786832 is associated with *mathematical ability*, rs2930456 is associated with *feeling nervous* and rs12359689 is associated with *feeling miserable*. These data indicate that multiple phenotypes have more than one LD-independent associated SNP at *SORCS3*. For example, five of the eight tag SNPs are associated with *mathematical ability* and three of the eight tag SNPs are associated with *unipolar depression*.

### 3.2. Functional Annotation of Associated SNPS at SORCS3

Analysis using HaploReg did not identify any of the eight tag SNPs to be in LD (*r*^2^ > 0.2) with missense or loss-of-function variants in *SORCS3*. Analysis of eQTL data for each tag SNP and their linked SNPs in each haplotype block identified a consistent pattern of results. For all the groups of tag SNPs, there is at least one example of an SNP where the allele associated with more favourable outcomes across the phenotypes (e.g., decreased risk of a neuropsychiatric disorder or better cognitive ability) is also associated with increased *SORCS3* expression in various tissue types, including brain regions (Appendix A).

### 3.3. Development of SORCS3 Protein Interaction Gene-Set

Using SORCS3 as bait, we found that SORCS3 has multiple direct (1st degree; n = 14) and indirect (2nd and 3rd degree; n = 19) protein interactors in the STRING, BioGRID, IntACT and HPRD databases. We then used each of the fourteen 1st degree protein interactors as bait to search for their interactors in each database. This identified an additional 378 indirect (2nd degree) protein interactors of SORCS3 and, in total, resulted in a final SORCS3 protein interaction gene-set of 412 genes (Appendix A).

### 3.4. Heritability Analysis of SORCS3 Gene-Set

sLDSC analysis was performed to test if the SORCS3 protein-interacting gene-set was enriched for heritability that contributes to single phenotypes with SNP association signals at *SORCS3*. Enrichment values were considered significant at a Bonferroni-corrected *p*-value of <0.05. Thirteen test phenotypes and five control phenotypes were analysed. When the start/stop regions of each gene were found, the average size of each genomic region was 86 kb. In total, the 412 SORCS3 protein-interacting genes accounted for 3.72% of 1,217,311 SNPs in the sLDSC analysis. The SORCS3 gene-set was significantly enriched for heritability contributing to SCZ, BPD, IQ and EA after multiple test correction (Figure 3). Results for the control phenotypes were non-significant. Full sLDSC results are in Appendix A.

### 3.5. Individual Gene Analysis Using MAGMA

For the four disorders and traits with enriched heritability, we used MAGMA v1.09 to identify individual genes within the SORCS3 gene-set that were associated at genome-wide significant levels. Three genes were associated with BPD, 20 with EA, 14 with IQ and 26 with SCZ (Appendix A). Eleven genes were associated with more than one phenotype, but other than *SORCS3*, only *RBFOX1* was associated with three phenotypes (EA, IQ and SCZ).

### 3.6. Functional Annotation of SORCS3 Gene-Set

In total, 61 of 412 genes in the gene-set were mapped to 60 unique SynGO annotated genes; 43 genes had a Cellular Component (CC) annotation and 50 genes had a Biological Processes (BPs) annotation. Five CC terms and 15 BP terms were significantly enriched at a false discovery rate (FDR) Q-value < 0.01 (Figure 4). The CC terms were in the postsynapse, including the postsynaptic membrane and postsynaptic density. The postsynaptic density term (GO:0014069) includes 14 genes from the SORCS3 gene-set, of which three genes in addition to *SORCS3* are individually associated with phenotypes that exhibited significantly enriched heritability (*CTNNB1* (IQ), *SHANK2* (BPD), *SHANK3* (IQ and EA)). The BP terms were clustered within three sections of the hierarchy: the process in the postsynapse, synapse organisation and trans-synaptic signalling. The trans-synaptic signalling term (GO:0099537) includes 19 genes from the SORCS3 gene-set, of which four genes in addition to SORCS3 are individually associated with phenotypes that exhibited significantly enriched heritability (*MAPK3* (SCZ), *SHANK3* (IQ and EA), *TENM2* (EA) and *TENM4* (IQ and EA)).

## 4. Discussion

Neuropsychiatric and neurodevelopmental brain-related disorders and traits that have an impact on the experience of feeling, emotion or mood or cognitive function are highly polygenic in nature, exhibiting a multitude of significantly associated genetic variants with small effect sizes. GWAS has identified many genes that contain variants associated with multiple brain-related phenotypes [23,36,54]. Here, we have shown *SORCS3* to be a striking example of a gene where the same SNP or its linked haplotype of high-LD variants is associated with multiple phenotypes and where multiple LD-independent SNPs are associated with the same phenotypes. Among these SNPs, the alleles associated with the more favourable outcome for each phenotype are all associated with increased expression of the *SORCS3* gene. This suggests that there may be a consistent functional effect for the LD-independent associated SNPs, but the multiple causative SNPs and the mechanism of their functional effect on *SORCS3* gene expression remain to be elucidated.

We used protein–protein interaction data to create a SORCS3 gene-set (n = 412 genes) that was enriched for SNP heritability for SCZ, BPD, IQ and EA. This indicates that *SORCS3* functions within a network of genes that contribute to psychiatric disorders and behavioural phenotypes. Given the established role of SORCS3 within the synapse, we specifically investigated the SORCS3 gene-set using the SynGO synaptic gene ontologies, which is an evidence-based, expert-curated resource for synapse function and gene enrichment studies. The enriched ontology terms to emerge from this analysis included the postsynaptic membrane, postsynaptic density and the process in postsynapse, synapse organisation and trans-synaptic signalling. This highlights the compartments and functions of the synapse, where genetic variation in *SORCS3* and the genes encoding its protein interactors likely contribute to the risk of neuropsychiatric illness and variations in cognitive ability within the population. Further analysis of gene-based results from GWAS and these synaptic gene ontologies identified a number of SORCS3 protein interactors as interesting contributors to brain-related disorders and traits.

*RBFOX1* encodes the RNA binding protein Fox-1 Homolog 1. It is also known as Ataxin-2 binding protein 1 (*A2BP1*) and is mainly expressed in neurons, the heart and muscles [55,56]. A recent study reported on an animal model of ASD, where brain-functional MRI studies revealed the association of the *RBFOX1* variant with enhanced neural activity to emotional stimuli, less prefrontal processing and increased fear expression [57]. It also has a critical role in the alternative splicing and stabilisation of transcripts that encode proteins essential for neurotransmission, revealed from cross-linking immunoprecipitation and RNA-Seq and functional studies in knockout mouse models [58,59,60]. Alternative splicing of *RBFOX1* results in nuclear and cytoplasmic isoforms. The nuclear isoform regulates splicing, while the cytoplasmic isoform plays a role in the stability of mRNA and promotes translation. The nuclear isoform is important for the control of neuronal excitation in the mammalian brain due to its involvement in neuron migration and synapse network formation within the cerebral cortex [61,62,63]. Along with a key role as A2BP1 in spinocerebellar ataxia type-2, the other structural variants and deletions in the *RBFOX1* gene increase the risk and development of disorders associated with aggression, generalised epilepsy, ASD, intellectual disability, and cognitive dysfunction in SCZ [64,65,66,67]. A recent study of *RBFOX1* as a candidate gene for aggressive behaviour has highlighted the contribution of *RBFOX1* to aggression and to other psychiatric and neurodevelopmental disorders that often manifest aggression. This study further highlighted variations in the *RBFOX1* gene influencing temporal lobe volume in AlzD at a genome-wide significance level, and several aggressive-related phenotypes have shown alteration in temporal lobe volume. *RBFOX1* was also related to aggression in Drosophila and mice models [65].

*CTNNB1* encodes the β-catenin protein, which is present in many cell types and tissues and is primarily located at junctions that connect neighbouring cells (adherens junctions). β-catenin also plays an important role in cell adhesion and intercellular communication and is involved in the Wnt signalling pathway during interneuron development. *CTNNB1* is important in the development and maturation of the brain, and de novo mutations in the gene can cause intellectual disability [68,69]. Recent findings reported a significant increase in the mRNA expression levels of *CTNNB1* in both SCZ and BD cohorts when compared with the control. This suggests that the expression of *CTNNB1* could be used as a possible biomarker in aiding the diagnosis of SCZ and BD. This study also found that a variant in the 3′-UTR of *CTNNB1,* rs2953, also influences the risk of SCZ in the Han Chinese population and modifies the binding of miR-485 to *CTNNB1* [70].

The *SHANK* gene family, *SHANK1, SHANK2* and *SHANK3*, encode a set of scaffolding proteins enriched in the postsynaptic density of excitatory synapses. Increasing evidence supports a role for the *SHANK* family in a wide range of neuropsychiatric disorders. SHANK proteins contain multiple domains for protein–protein interaction, including ankyrin repeats and an SH3 domain. Genetic defects in *SHANK2* are linked to a variety of neuropsychiatric disorders, including ASD, SCZ, and BPD. For instance, rare genetic variants in both *SHANK2* and *SHANK3* have been described in individuals presenting with primary diagnoses of SCZ or BPD [71]. The SHANK3 protein is also involved in the formation and maturation of dendritic spines. SHANK3 also acts as a synaptic skeleton and helps to maintain synaptic transmission and plasticity [72] SHANK3 is expressed at high levels in the developing neurons of the cerebral cortex and cerebellum. Mutations in mouse *Shank3* lead to a number of behavioural changes, including an increase in repetitive routines, altered social behaviour and anxiety-like phenotypes [73]. These changes are comparable to behavioural abnormalities seen in ASD. Copy number mutations of the *SHANK3* gene can lead to Phelan–McDermid syndrome, which is a rare genetic disorder characterised by a number of developmental abnormalities, including intellectual disability, poor motor tone, and ASD-like symptomology [74].

*MAPK3* is a member of the *MAP* kinase family, also known as extracellular signal-regulated kinases (ERKs). MAPK3 acts in response to extracellular signalling, leading to cell proliferation and the differentiation and progression of the cell cycle. *MAPK3* also plays a central role in regulating the precise targeting of pre-synaptic axons to proper postsynaptic targets. The appropriate pairing of post- and pre-synaptic partners in neuronal circuits is critical for proper functionality. Errors induced in this process may play a role in the development of neuropsychiatric disorders [75]. A mutation of the *rolled* gene in Drosophila, a homolog of *MAPK3*, causes the ectopic innervation of axonal branches [75]. *MAPKs* have also been found to play a role in synaptic plasticity, learning, and memory. Disruption of these signalling pathways can lead to pathological function and has been implicated in degenerative conditions such as AlzD, Parkinson’s disease and amyotrophic lateral sclerosis, as well as psychiatric disorders of MDD, BPD, ASD and SCZ [76].

*TENM2* (teneurin-2), encoding a member of the teneurin protein family, is an evolutionarily conserved adhesion molecule that is highly expressed in neurons but is also present in other tissues [77]. *TENM2* is a type II transmembrane protein containing a very large extracellular sequence, an N-terminal cytoplasmic sequence, and a single transmembrane region [78]. *TENM2* has been shown to interact with latrophilin-1, thus establishing its role in intracellular calcium signalling. Latrophilin-1 is a neuronal G-protein-coupled receptor implicated in the control of Ca2+ and neurotransmitter release [79]. *TENM2* is critical for neuronal migration [80]. In a recent familial whole exome sequencing study of SCZ, *TENM2* emerged as a candidate gene harbouring private or rare damaging variants in multiple affected individuals [81]. *TENM4* is implicated in neuronal plasticity and signalling. Altered expression of *Ten-m* leads to a number of behavioural changes in animal models, including lower learning ability, sleep reduction, and increased aggressiveness [82]. RNA sequencing in *Drosophila* brain tissue revealed that abnormal *Ten-m* expression led to gene expression changes related to neurogenesis and ATPase activity [83].

In terms of study limitations, it is relevant to note that none of these proteins are direct interactors of SORCS3, and molecular experimental work will be required to establish their functional relationship with SORCS3 in different brain regions and cell types. These genes do represent, similar to *SORCS3*, genes associated with the risk of neuropsychiatric disorders and behavioural phenotypes that have multiple neuronal and synaptic functions. However, none surpass *SORCS3* in terms of showing association with as many phenotypes and containing multiple LD-independent SNPs associated with the same phenotypes. This highlights SORCS3 as an important gene for ongoing studies of the molecular mechanisms of brain-related disorders and traits.

## Figures and Tables

**Figure 1 genes-14-00482-f001:**
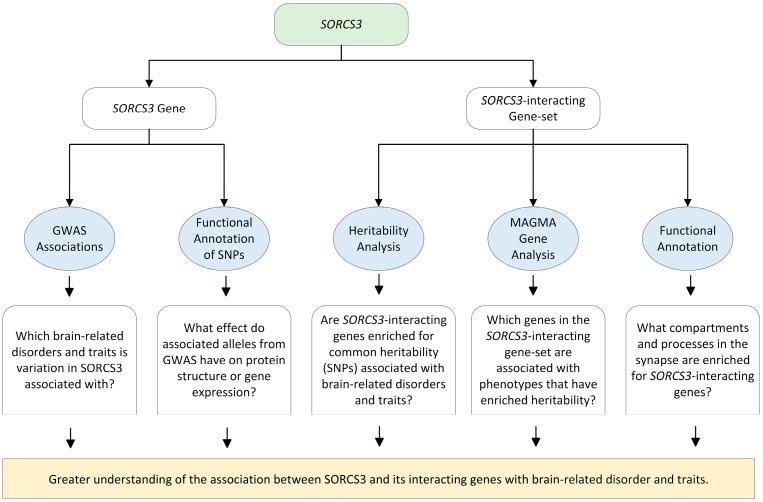
Workflow of the study design employed to ask multiple research questions related to SORCS3′s association with brain-related disorders and traits.

**Figure 2 genes-14-00482-f002:**
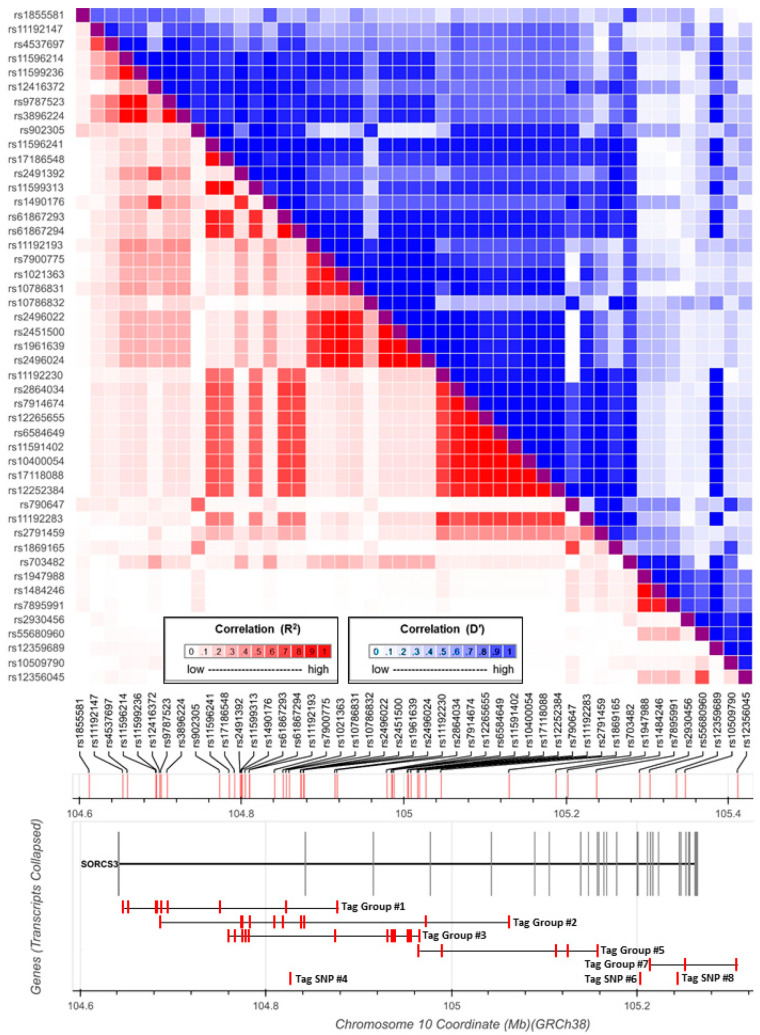
Plot of the SORCS3 gene. The LD relationship between the 46 SNPs is displayed in measures of *r*^2^ (red, below the diagonal) and *D’* (blue, above the diagonal). SNPs are mapped to their position along the gene and to their tag groups, which are high-LD blocks containing multiple associated SNPs. Three SNPs are not in LD with other SNPs (rs10786832 (tag SNP #4), rs2930456 (tag SNP #6) and rs12359689 (tag SNP #8)). rs1855581 at the 5′ end of the gene is not an associated SNP and is just present to extend the map to include exon 1 and, thus, display all exons of the gene.

**Figure 3 genes-14-00482-f003:**
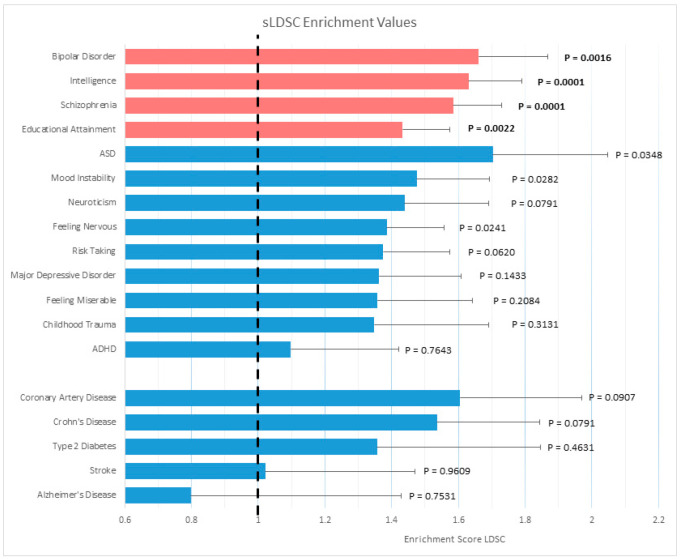
Heritability enrichment analysis of the SORCS3 gene-set stratified linkage disequilibrium score regression (sLDSC). Test (top) and control (bottom) phenotypes are plotted on the y-axis. Enrichment values (proportion of h2/proportion of # SNPs), where an enrichment score of 1 indicates no enrichment, are plotted on the x-axis, with error bars representing standard error and p-values displayed. Phenotypes where the p-value remains significant after Bonferroni correction are shown in red. Full results of the sLDSC analysis are located in Appendix A.

**Figure 4 genes-14-00482-f004:**
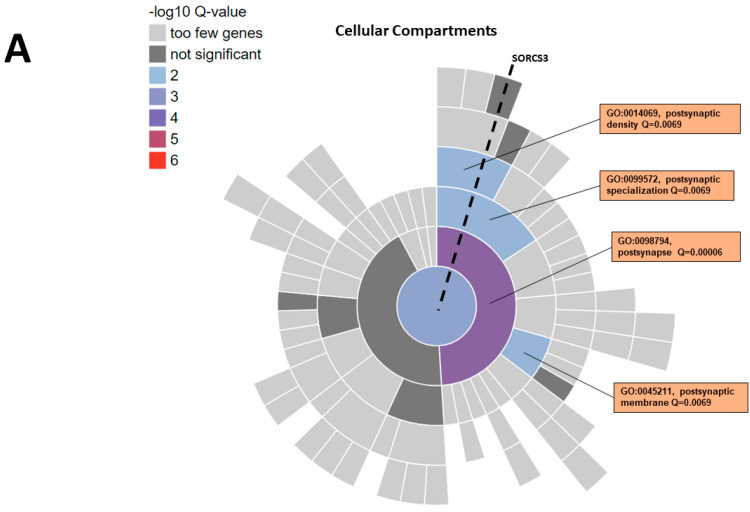
SynGO enrichment of the SORCS3 gene-set. The central circle in the sunburst plot includes all SynGO annotated genes, which then subdivides based on the hierarchy of annotation terms. The dashed line runs through the SynGO terms that contain SORCS3. (**A**) Five Cellular Component terms are significantly enriched at 1% FDR (testing terms with at least three matching input genes), and these terms are in the postsynapse. (**B**) Fifteen Biological Processes terms are significantly enriched, and these terms are in synapse organisation (blue terms), postsynaptic processes (orange terms) and synaptic signalling (red terms).

## Data Availability

Not applicable.

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
