# Peer review of "Independent Associated SNPs at SORCS3 and Its Protein Interactors for Multiple Brain-Related Disorders and Traits"

_genes, 2023, doi:10.3390/genes14020482_

Round 1

Reviewer 1 Report

The manuscript by Kamran et al., is a meta-analysis paper where they tried to extract various association between SORCS3 and brain-related disorders. In general, the manuscript is well written and could be useful for people in the field. Following are my comments:

1.     The analysis based on String database needs to have a figure in the form of a network diagram.

2.     How do they measure the strength of interactions and how statistically relevant are these hits? Are they based on experiments or coexpression? An additional table with these details should be provided. 

3.     Could they predict a pathway for SORCS3 and its involvement in various brain disorders mentioned in the manuscript?

4.     Do they see any overlap with genes identified by Wu et al.2020, other than SORCS3?

5.     I don’t see sections for Data and code availability and Author contributions in the manuscript.

Author Response

The manuscript by Kamran et al., is a meta-analysis paper where they tried to extract various association between SORCS3 and brain-related disorders. In general, the manuscript is well written and could be useful for people in the field.

Response:

  • The analysis based on String database needs to have a figure in the form of a network diagram.

Response:

We have not included a figure of the STRING analysis for the following reasons: (i) The generation of the SORCS3 gene-set based on protein-protein interactions utilized String plus other databases detailing protein-protein interactions (BioGRID, IntACT and HPRD). Not all databases provide an application to generate figures that capture the reported protein-protein interactions. Therefore, we are unable to provide a figure that captures the entire gene-set. (ii) We provide a link to the STRING database and any researcher can quickly create the String-only interactive figure online by searching for SORCS3.

  • How do they measure the strength of interactions and how statistically relevant are these hits? Are they based on experiments or coexpression? An additional table with these details should be provided.

Response:

These interactions are based on a mixture of primary experimental research, physical interactions from high-throughput assays and other methods such as text-mining. Databases like STRING generate a composite metric of evidence including all of these sources (if applicable). To ensure consistency and ensure a thorough network was built, default settings were used for all databases not excluding any forms of evidence. To address this comment, we have updated Supplementary Table 2 to include the evidence source for each of the 1st degree interactors of SORCS3.

  • Could they predict a pathway for SORCS3 and its involvement in various brain disorders mentioned in the manuscript?

Response:

At this stage we do not have enough evidence to predict a specific pathway. However, our gene ontology analysis using SynGO shows enrichments in multiple synaptic ontologies, which help predict where in neuronal/synaptic function a specific pathway may is exist that is perturbed by SORCS3 dysfunction leading to increased risk of psychiatric disorders. These includes enriched cellular compartments such as postsynaptic membrane (GO:0045211) and postsynaptic density (GO:0014069) as well as biological processes such as trans-synaptic signalling (GO:0099537) as being potentially involved.

  • Do they see any overlap with genes identified by Wu et al. 2020, other than SORCS3?

Response:

SORCS3 is the only gene associated with all 5 phenotypes by Wu et al. 2020. No other genes highlighted in 4+ phenotypes appear in the SORCS3 gene set. Among genes overlapping in 3+ phenotypes, an additional gene, ABT1 (BD_mtag & DEP_mtag & SCZ_mtag), is shared with the SORCS3 gene set.

  • I don’t see sections for Data and code availability and Author contributions in the manuscript

Response:

We have now added two sections “Data Availability Statement” and “Author Contributions” to address this comment. All analysis involving code (MAGMA and sLDSC) are described within Materials and Methods section.

Reviewer 2 Report

In this study, authors have utilized publicly available databases to analyze, identify and catalog the SNPs present in SORCS3 that are associated with multiple brain-related disorders and traits. Through multiple bioinformatics analyses, they have observed SORCS3 SNPs (which affects its protein expression) and their indirect interactors (RBFOX1, CTNNB1, SHANK family genes, MAPK3, and TENM2) play an important role in multiple brain-related disorders and traits.

However, all these are based on bioinformatics analysis. Authors in the discussion have cited multiple references in support of these observations. This manuscript would have been more substantial if it had been supported by some experimental data (like validation of SORCS3 levels in patient samples with brain-related disorders and traits). Authors have described a few indirect protein interactors playing an important role along with SORCS3 in developing brain-related diseases. They could have explored the different direct and indirect interactors of SORCS3, which might play an essential role in regulating SORCS3 and thus contribute to the development of multiple brain-related disorders and traits.

In the results section “Heritability analysis of SORCS3 gene-set” paragraph line 4, it is mentioned as twelve test phenotypes, but in Figure 2 and supplementary table 3, there are 13 test phenotypes. So, is it twelve or 13 test phenotypes?

Author Response

In this study, authors have utilized publicly available databases to analyze, identify and catalog the SNPs present in SORCS3 that are associated with multiple brain-related disorders and traits. Through multiple bioinformatics analyses, they have observed SORCS3 SNPs (which affects its protein expression) and their indirect interactors (RBFOX1, CTNNB1, SHANK family genes, MAPK3, and TENM2) play an important role in multiple brain-related disorders and traits.

2.1 However, all these are based on bioinformatics analysis. Authors in the discussion have cited multiple references in support of these observations. This manuscript would have been more substantial if it had been supported by some experimental data (like validation of SORCS3 levels in patient samples with brain-related disorders and traits).

Response:

Although we agree that it would be interesting to have further experimental data to support these findings, that was beyond the scope of this study. Our study uses bioinformatic analyses to come to novel conclusions by combining existing data from multiple sources that have not been analysed together previously. This serves to make efficient use of currently available datasets. We have edited our discussion to note the requirement for future molecular experimental to establish the functional relationship of interacting associated genes with SORCS3 in different brains regions and cell types.

2.2 Authors have described a few indirect protein interactors playing an important role along with SORCS3 in developing brain-related diseases. They could have explored the different direct and indirect interactors of SORCS3, which might play an essential role in regulating SORCS3 and thus contribute to the development of multiple brain-related disorders and traits.

Response:

We believe our study does do this but using the bioinformatic tools and datasets that we had available. As stated about, molecular experimental work will be required to establish their functional relationship with SORCS3 in different brains regions and cell types.

2.3 In the results section “Heritability analysis of SORCS3 gene-set” paragraph line 4, it is mentioned as twelve test phenotypes, but in Figure 2 and supplementary table 3, there are 13 test phenotypes. So, is it twelve or 13 test phenotypes?

Response:

Thank you for this observation. This was an error as it is 13 test phenotypes. This has now been corrected in the text.

Reviewer 3 Report

Comments/Suggestions

1. Abstract: "overlap with to synaptic biology", wither change the sentence or remove "to"

2.  "As these studies [30] have identified up to hundreds of loci containing thousands of genes, only a small proportion of associated genes are highlighted and discussed in the manuscripts describing these studies" 

This citation is a self-citation, and it was about the "major depression disorder" and I do not see any relation to the citation here.

3. Also, you have cited your paper in the opening of the discussion section in which it was mentioned that many genes associated with many brain-related phenotypes. In addition to your citation please also include the Howard et al. (36) and Wu et al. (23).

4. Expression quantitative trait loci data (eQTL) were sourced from the Genotype-Tissue Expression (GTEx) project (https://gtexportal.org). 

Please reference to this sentence

5. Sometimes author used small p and capital P for p values, please use p in the entire text. 

6. Page 8. "We Bonferroni corrected", please rephrase the sentence

7. Results: Figure2 caption: it was mentioned the phenotype that has p value less than 0.05 exhibited in red color, however, there are other disease phenotypes such mood instability, ASD, and Feeling nervous are also have p value less than 0.05 which are still represented with blue color. Please change the figure or change the caption. 

8. In the following link: "https://www.genecards.org/cgi-bin/carddisp.pl?gene=SORCS3" it was mentioned that SORCS3 is associated with Alzheimer's disease, however, in your data it is not associated with AD. would you please elaborate why you found contradictory results in your data.

9. Page 12: "Across these SNPs, alleles" SNPs and alleles

10. Discussion section needed to be rewritten. Authors have discussed many other gene types and their expression in different disease. In the discussion section, you must emphasize how your findings are associated or contradicting your peers' findings rather than brief explanation of other gene type. 

Author Response

3.1 Abstract: "overlap with to synaptic biology", wither change the sentence or remove "to"

Response:

Thank you for this observation. This has now been corrected in the text.

3.2  "As these studies [30] have identified up to hundreds of loci containing thousands of genes, only a small proportion of associated genes are highlighted and discussed in the manuscripts describing these studies"

This citation is a self-citation, and it was about the "major depression disorder" and I do not see any relation to the citation here.

Response:

Upon reflection, we agree that this citation may not be appropriate in this circumstance and it has been removed.

3.3 Also, you have cited your paper in the opening of the discussion section in which it was mentioned that many genes associated with many brain-related phenotypes. In addition to your citation please also include the Howard et al. (36) and Wu et al. (23).

Response:

Thank you for this suggestion. These references have now been added.

3.4 Expression quantitative trait loci data (eQTL) were sourced from the Genotype-Tissue Expression (GTEx) project (https://gtexportal.org).

Please reference to this sentence

Response:

Thank you for this comment. This sentence has now been appropriately referenced.

3.5 Sometimes author used small p and capital P for p values, please use p in the entire text.

Response:

Thank you for this observation. Instances of capital P being used for p-value in the manuscript have been changed.

3.6 Page 8. "We Bonferroni corrected", please rephrase the sentence

Response:

Thank you for this observation. This sentence has now been rephrased.

3.7 Results: Figure2 caption: it was mentioned the phenotype that has p value less than 0.05 exhibited in red color, however, there are other disease phenotypes such mood instability, ASD, and Feeling nervous are also have p value less than 0.05 which are still represented with blue color. Please change the figure or change the caption.

Response:

Only phenotypes where the p-value remains significant after Bonferroni correction are shown in red. The figure legend text has been edited to clarify this point.

3.8 In the following link: "https://www.genecards.org/cgi-bin/carddisp.pl?gene=SORCS3" it was mentioned that SORCS3 is associated with Alzheimer's disease, however, in your data it is not associated with AD. would you please elaborate why you found contradictory results in your data.

Response:

Genecards states that “candidate gene studies suggest that genetic variation in this gene is associated with Alzheimer's disease”. However, subsequent GWAS do not support this “suggestion”. Our review of GWAS Catalog only identifies one associated SNP for Alzheimer's disease (the actual reported trait is age of onset of the disease) but the p-value does not reach genome-wide significance levels. Therefore, we believe our results are consistent with published genetic studies and it was appropriate to use Alzheimer's disease is a control brain disorder to assess the validity of our analyses of our test brain disorders and traits.

3.9 Page 12: "Across these SNPs, alleles" SNPs and alleles

Response:

Thank you for this observation. “SNPs and alleles” does not capture what we wanted to discuss, however we agree the sentence may be slightly unclear. We have changed the phrasing of the sentence for clarity.

3.10 Discussion section needed to be rewritten. Authors have discussed many other gene types and their expression in different disease. In the discussion section, you must emphasize how your findings are associated or contradicting your peers' findings rather than brief explanation of other gene type.

Response:

We disagree that the Discussion as a whole must be rewritten. Our intent was to report our overall findings from a functional aspect and briefly discuss some genes we found to be most relevant from our analyses. In terms of our findings associating with or contradicting our peers' findings, which is certainly important, our paper is distinct from previous studies in that it is the first to our knowledge to pull together all available GWAS data for SORCS3 and to develop a set of interacting genes based on protein-protein interaction data. Therefore, we cannot directly relate our study to previous studies.

Reviewer 4 Report

The manuscript “Independent associated SNPs at SORCS3 and its protein interactors for multiple brain-related disorders and traits” by Muhammad Kamran et al. showed the independent association signals at SORCS3 with brain-related disorders and traits, with the effect possibly mediated by reduced gene expression resulting in an impact on synaptic function. In addition, the authors conducted a systematic search of previously published genome-wide association studies to identify and catalog associations between SORCS3 and brain-related disorders and traits. However, the manuscript must be revised to meet the journal’s publication standards. The following are some key points:

Major recommendations

1.       Describe the Brain-related disorders and traits they want precisely throughout the manuscript, currently, it sounds very vague.

2.       Describe the process of selecting the SORCS3 gene set (n=412 genes) for the sake of reproducibility.

3.       Please provide a schematic diagram of the workflow to convey the study’s rationale to a large audience and readers.

Minor remarks

1.       All the Figure pixels should be improved. Figures 3A and 3B should be combined or the authors should separate the Figure legend.

2.       The introduction is lacking some recent references pertaining to SorCS3 promoting the internalization of p75NTR to inhibit GBM progression.

3.       The discussion should be updated by discussing various disorders and traits which are under the control of SORCS3.

4.       Methodology subsections should be elaborated in detail for the ease of reproducibility by others, as well as some backup references for the analysis.

Author Response

The manuscript “Independent associated SNPs at SORCS3 and its protein interactors for multiple brain-related disorders and traits” by Muhammad Kamran et al. showed the independent association signals at SORCS3 with brain-related disorders and traits, with the effect possibly mediated by reduced gene expression resulting in an impact on synaptic function. In addition, the authors conducted a systematic search of previously published genome-wide association studies to identify and catalog associations between SORCS3 and brain-related disorders and traits. However, the manuscript must be revised to meet the journal’s publication standards. The following are some key points:

Major recommendations

4.1 Describe the Brain-related disorders and traits they want precisely throughout the manuscript, currently, it sounds very vague.

Response:

We have now more clearly defined brain-related disorders and traits in our Abstract, Introduction and Discussion. In the Introduction we state: “In this study, we consider brain-related disorders that are neuropsychiatric or neurodevelopmental and traits that have an impact on experience of feeling, emotion or mood or cognitive function.”

4.2 Describe the process of selecting the SORCS3 gene set (n=412 genes) for the sake of reproducibility.

Response:

Further detail has been added to the Materials and Methods section to clarify how the gene-set was generated.

4.3 Please provide a schematic diagram of the workflow to convey the study’s rationale to a large audience and readers.

Response:

Thank you for this suggestion. We agree this would more easily convey the study’s rationale and workflow. A new Figure 1 workflow has now been included to detail the study design employed to ask multiple research questions related to SORCS3’s association with brain-related disorders and traits.

Minor remarks

4.4 All the Figure pixels should be improved. Figures 3A and 3B should be combined or the authors should separate the Figure legend.

Response:

Thank you for this observation. We will consult the editor regarding the quality and number of figures.

4.5 The introduction is lacking some recent references pertaining to SorCS3 promoting the internalization of p75NTR to inhibit GBM progression.

Response:

This is an interesting paper. However, malignancies of neuronal or glial cells in the brain such as GBM do not fit into the scope of our study and we have limited our review of the literature to the role of SORCS3 in disorders and traits as described in our response to 4.1 above.

4.6 The discussion should be updated by discussing various disorders and traits which are under the control of SORCS3.

Response:

We have reviewed the Discussion and following on from the previous point, we have restricted our Discussion to just SORCS3 and interacting genes associated neuropsychiatric and neurodevelopmental brain-related disorders and traits that have an impact on experience of feeling, emotion or mood or cognitive function.

4.7 Methodology subsections should be elaborated in detail for the ease of reproducibility by others, as well as some backup references for the analysis.

Response:

We have reviewed and edited the Materials and Methods to make sure that all activities have been described accurately and that weblinks and references are in place to direct researchers to the appropriate resources to enable reproduction of this work.

Round 2

Reviewer 3 Report

I appreciate authors efforts to address my comments. However, authors disagree to change the discussion section. I do understand why authors would like to discuss other types of relevant genes. However, there is no flow in the discussion section. Each paragraph is individual paragraph without connection.  Moreover, there is not much discussion around how other genes are relevant to SORCS3. The manuscript would improve significantly if authors reconsider to change the discussion section.